# Position: Agentic AI System Is a Foreseeable Pathway to AGI

**Junwei Liao** [1 2]   **Shuai Li** [1 2]   **Muning Wen** [1]   **Jun Wang** [3]   **Weinan Zhang** [1 2]

## Abstract

Is monolithic scaling the only path to AGI? This paper challenges the dogma that purely scaling a single model is sufficient to achieve Artificial General Intelligence. Instead, we identify Agentic AI as a necessary paradigm for mastering the complex, heterogeneous distribution of real-world tasks. Through rigorous theoretical derivations, we contrast the optimization constraints of monolithic learners against the efficiency of Agentic systems, progressing from simple routing mechanisms to general Directed Acyclic Graph (DAG) topologies. We demonstrate that Agentic AI achieves exponentially superior generalization and sample efficiency. Finally, we discuss the connection to Mixture-of-Experts, reinterpret the instability of current multi-agent frameworks, and call for greater research focus on Agentic AI.

## 1. Introduction

The No Free Lunch Theorem (Wolpert & Macready, 1997) dictates that no universal intelligence can perform perfectly on every conceivable task. Consequently, given the inductive nature of real-world problems, the objective is to achieve AGI within the context of the human world. But how is AGI defined in this sense? Historically, Machine Intelligence has been subject to numerous interpretations (Gudwin, 2000; Horst, 2002). Legg and Hutter, after surveying various perspectives, define it as an agent's ability to *"achieve goals in a wide range of environments,"* which aligns with most definitions (Legg & Hutter, 2007). Furthermore, Chollet posits that *"the intelligence of a system is a measure of its skill-acquisition efficiency over a scope of tasks, with respect to priors, experience, and generalization difficulty"* (Chollet, 2019). **In essence, within the scope of our physical existence, AGI necessitates optimal performance across a near-infinite spectrum of human-relevant tasks.**

Reed et al. (2022) states that *"...such an agent (which is generally capable on a large number of tasks) can be obtained through scaling data, compute and model parameters, continually broadening the training distribution while maintaining performance..."*

Despite relentless scaling of data and computation, no single monolithic model commands ubiquitous dominance across all benchmarks (Jimenez et al., 2024; Mialon et al., 2024; Patil et al., 2025; Phan et al., 2025), and the elusive quality of true AGI has notably failed to emerge despite the saturation of high scores. While scaling pushes performance boundaries, it yields diminishing returns at prohibitive costs (Kaplan et al., 2020; Hoffmann et al., 2022; Pearce & Song, 2024; Porian et al., 2024), resulting in narrow proficiency peaks rather than superiority across the full spectrum of real-world tasks. This limitation stems from strong biases introduced by specific optimization objectives and training data (Battaglia et al., 2018), a problem that is exacerbated when synthetic data is employed (Dohmatob et al., 2024).

The term **Agentic AI is formally proposed as a paradigm marked by multi-agent collaboration, dynamic task decomposition, and coordinated autonomy** (Sapkota et al., 2026). From isolated to coordinated, Agentic AI moves beyond monolithic scaling to bring up more aspects of orchestrating the multi-agent systems. Actually, platforms like Manus AI (Manus, 2024) and coding assistants such as Codex (OpenAI, 2024), Claude Code (Anthropic, 2024), have preliminarily exemplified the power of Agentic AI, while the paradigm is rapidly expanding into diverse domains such as information retrieval (Zhang et al., 2025b) and standardized inter-agent communication protocols (Yang et al., 2025). However, most AI research centers on monolithic models, and there is still no concrete theoretical proof showing that Agentic AI is overall superior to the monolithic approach.

In this work, we present a series of demonstrations and theoretical derivations to substantiate the claim that **Agentic AI is the foreseeable cross-level move towards AGI**. This capability arises from its ability to adaptively decompose tasks into correlated atomic ones and orchestrate specific agents with distinct biases, thereby aligning with real-world structures and pushing Pareto optimality. The remainder of

---

[1]Shanghai Jiao Tong University [2]Shanghai Innovation Institute [3]University College London. Correspondence to: Weinan Zhang <wnzhang@sjtu.edu.cn>.

*Proceedings of the $43^{rd}$ International Conference on Machine Learning*, Seoul, South Korea. PMLR 306, 2026. Copyright 2026 by the author(s).

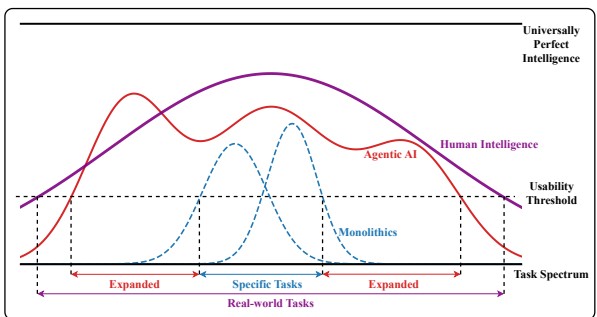

*Figure 1.* Agentic AI expands the range of usable tasks and improves performance compared to monolithic models. While monolithic models exhibit narrow performance peaks only on specific tasks they are trained for, Agentic AI demonstrates multi-peak performance across a broader spectrum. This expands usable capabilities, approaching and even surpassing the altitude and breadth of human intelligence.

the paper is organized as follows: In Section 2, we establish the theoretical foundations necessary for our proof by reviewing constraints from learning theory. In Section 3, we demonstrate the inability of monolithic models to achieve multi-peak performance and derive the advantage of routing-based Agentic AI. We then extend this analysis in Section 4 to general Agentic AI represented as directed acyclic graphs (DAGs) of Agents. We also list some alternative views in Section 5 and reinterpret them after conveying the main idea of the paper. Finally, in Section 7, we conclude by positioning Agentic AI as the inevitable successor to monolithic scaling on the path to AGI.

## 2. Theoretical Foundations

### 2.1. Structured Real-World Distribution

The No Free Lunch Theorem asserts that, without prior assumptions on the data distribution, no learning algorithm outperforms any other on average. However, real-world tasks are not uniform noise; they obey specific physical and semantic constraints. To rigorously analyze the advantage of Agentic AI, we formalize the data-generating process not merely as a statistical mixture, but as a collection of functions supported on low-dimensional manifolds.

**Definition 2.1** (Structured Real-World Distribution)**.** Let the input space be $\mathcal{X} \subseteq \mathbb{R}^D$ and the output space be $\mathcal{Y} \subseteq \mathbb{R}$. We define the *Structured Real-World Distribution* $\mathcal{D}_{\text{real}}$ as a measure on $\mathcal{X} \times \mathcal{Y}$ generated by a latent task variable $z \in \{1, \ldots, K\}$ with prior probabilities $\alpha_k = P(z = k)$. The joint distribution is defined by the tuple $(\mathcal{M}, \mathcal{F}, \boldsymbol{\alpha})$, characterized by the following structural properties:

1. Union of Manifolds: The support of the marginal distribution $P(x)$ is a union of $K$ distinct, compact Riemannian manifolds $\{\mathcal{M}_k\}_{k=1}^K$, where each $\mathcal{M}_k \subset \mathbb{R}^D$ has an intrin-

sic dimension $d_k \ll D$:

$$\text{supp}(P(x)) \subseteq \bigcup_{k=1}^K \mathcal{M}_k$$

2. Local Functional Consistency: For each task $k$, there exists a distinct labeling function $f_k : \mathcal{M}_k \to \mathcal{Y}$ such that the conditional distribution $P(y|x, z = k)$ is concentrated around $f_k(x)$ with noise $\xi$:

$$y = f_k(\text{Proj}_{\mathcal{M}_k}(x)) + \xi, \quad \text{where } x \in \mathcal{M}_k$$

3. Task Divergence: The optimal functions are heterogeneous, meaning for any pair $j \neq k$, the functional distance implies distinct optimization landscapes:

$$\inf_{\theta \in \Theta} \mathbb{E}_{x \sim \mathcal{M}_k}[\ell(h_\theta(x), f_k(x))] \neq \inf_{\theta \in \Theta} \mathbb{E}_{x \sim \mathcal{M}_j}[\ell(h_\theta(x), f_j(x))]$$

Consequently, the density of the structured distribution is given by:

$$\mathcal{D}_{\text{real}}(x, y) = \sum_{k=1}^K \alpha_k \cdot \mathbb{I}_{\mathcal{M}_k}(x) \cdot P(y|f_k(x))$$

This definition elevates the premise from a simple probabilistic mixture to a piecewise-smooth manifold learning problem.

### 2.2. Theorems on Generalization Bounds

The Curse of Dimensionality (Bellman et al., 1957) creates volumetric sparsity as dimension $D$ increases. This is illustrated by the vanishing ratio of a hypersphere's volume to its enclosing hypercube:

$$\lim_{D \to \infty} \frac{V_{\text{sphere}}(r, D)}{V_{\text{cube}}(r, D)} = 0$$

Consequently, high-dimensional data concentrates in the domain's "corners". This increases the average distance between nearest neighbors, rendering local density estimation intractable.

Due to the volumetric sparsity discussed above, covering the domain $\Omega$ sufficiently to ensure small $\|x - x'\|_2$ requires a sample size $N$ that grows exponentially with $D$. This limitation is formally quantified by the minimax lower bound.

**Proposition 2.2** (Minimax Lower Bound on Compact Domains (Stone, 1982))**.** *Let $\mathcal{F}_L(\Omega)$ be the class of $L$-Lipschitz functions restricted to a compact subset $\Omega \subset \mathbb{R}^D$. Under the standard non-parametric regression model, the minimax risk for any estimator $\hat{f}_N$ based on $N$ samples satisfies:*

$$\inf_{\hat{f}_N} \sup_{f \in \mathcal{F}_L} \mathbb{E}\left[ \int_\Omega |\hat{f}_N(x) - f(x)|\, dP(x) \right] \geq C \cdot N^{-\frac{1}{2+D}}$$

*where $P(x)$ is the marginal distribution of inputs supported on $\Omega$, and $C > 0$ is a constant independent of $N$.*

The term $N^{-\frac{1}{2+D}}$ reflects the curse: to maintain a fixed error level, $N$ must scale exponentially with $D$, mirroring the geometric expansion of the volume.

Recent theoretical advancements provide a rigorous foundation for understanding the efficiency of Transformer-based architectures. While Yun et al. (2020) established that Transformers are universal approximators capable of implementing precise contextual mappings, Jiang & Li (2024) advanced this further by deriving explicit Jackson-type approximation rates. They proved that the generalization error is intrinsically governed by the spectral decay properties of the target function's temporal coupling, represented by the singular value decay rate $\alpha$ of the attention mechanism.

By linking these spectral properties to the model's capacity, we can express the approximation error $\mathcal{E}$ as a function of the parameter count $P$ and the task's intrinsic dimension $d$. Under the standard architectural assumption that parameters scale quadratically with the hidden dimension ($P \propto m_h^2$)(Hoffmann et al., 2022) and the spectral theoretical observation that the decay rate $\alpha$ scales inversely with dimension ($\alpha \propto 1/d$), the approximation error follows a dimensionality-dependent power law:

$$\mathcal{E}(P) \approx C \cdot P^{-\frac{\kappa}{d}} \tag{1}$$

where $C$ is a task-dependent constant and $\kappa$ represents the regularity (smoothness) of the target function.

### 2.3. Multi-Class Learning

Since Agentic AI may involve routing problems, specifically, choosing a proper agent for a specific input, we introduce some multi-class learning theories. Let $\mathcal{X}$ be the instance space and $\mathcal{Y} = \{1, \ldots, K\}$ be the label space with $K$ classes. We consider a hypothesis class $\mathcal{H} \subseteq \{h : \mathcal{X} \to \mathcal{Y}\}$.

Natarajan Dimension (Natarajan, 1989) is the generalization of VC dimension (Vapnik & Chervonenkis, 1971) for multiclass classification problems (where the number of labels $K > 2$).

A set $S = \{x_1, \ldots, x_m\} \subseteq \mathcal{X}$ is Natarajan-shattered by $\mathcal{H}$ if there exist two "witness" functions $f_0, f_1 : S \to \mathcal{Y}$ such that $f_0(x_i) \neq f_1(x_i)$ for all $i$, and for any binary vector $\mathbf{b} \in \{0,1\}^m$, there exists $h \in \mathcal{H}$ such that:

$$h(x_i) = \begin{cases} f_0(x_i) & \text{if } b_i = 0 \\ f_1(x_i) & \text{if } b_i = 1 \end{cases}$$

The Natarajan Dimension $d_N(\mathcal{H})$ is the maximum size of such a shattered set.

Jin (2023) gave the upper bounds on the Natarajan dimension, $d_N(\mathcal{H})$, for the tree-based and neural network function classes as below.

**Theorem 2.3** (Natarajan Dimension Upper Bound for Tree-based Classifiers (Jin, 2023)). *Consider multi-class classification problems with $d$ classes and inputs in $\mathbb{R}^p$. Let $\Pi_{L,d}^{dtree}$ be the class of decision trees of depth $L$. Let $\Pi_{L,T,d}^{forest}$ be the class of random forests consisting of $T$ such decision trees. The Natarajan dimensions for these classes are upper bounded by:*

$$d_N(\Pi_{L,d}^{dtree}) = \mathcal{O}(L2^L \log(pd)),$$
$$d_N(\Pi_{L,T,d}^{forest}) = \mathcal{O}(LT2^L \log(pd)).$$

**Theorem 2.4** (Natarajan Dimension Upper Bound for Neural Network Classifiers (Jin, 2023)). *Let $\Pi_{p,S}^{\sigma}$ denote the class of feed-forward neural networks with a fixed structure $S$ and at most $p$ parameters for $d$-class classification. If the activation functions are restricted to binary or linear sets (denoted as $\Pi_{p,S}^{binary}$), or if the activation functions additionally include ReLU (denoted as $\Pi_{p,S}^{ReLU}$), then the Natarajan dimension for both cases is upper bounded by:*

$$d_N(\Pi_{p,S}^{binary}) = d_N(\Pi_{p,S}^{ReLU}) = \mathcal{O}(d \cdot p^2).$$

With the Natarajan dimension of a hypothesis class established, the relationship between model complexity and generalization performance can be characterized as follows.

**Theorem 2.5** (Generalization Error Bounds for Multiclass ERM (Daniely et al., 2011)). *For every hypothesis class $\mathcal{H}$ with a finite label set $\mathcal{Y}$, given a sample size $m$ and confidence parameter $\delta$:*

$$\epsilon_{\mathcal{H}}(m, \delta) \leq \epsilon_{ERM}(m, \delta) \leq O\left(\sqrt{\frac{d_N(\mathcal{H})\ln(|\mathcal{Y}|) + \ln(\frac{1}{\delta})}{m}}\right)$$

*where $\epsilon_{\mathcal{H}}(m, \delta)$ denotes the minimax (PAC) error achievable by the optimal learning algorithm, and $\epsilon_{ERM}(m, \delta)$ denotes the uniform ERM error, representing the worst-case guarantee for any Empirical Risk Minimizer.*

## 3. Why and How Much Monolithic Learner Falls Behind

In this section, we first provide a formal justification for the negative transfer phenomenon in monolithic models when facing heterogeneous tasks. We frame the Average Trap explicitly as the penalty for ignoring the modular structural bias of $\mathcal{D}_{\text{real}}$. Effectively, the monolithic model attempts to compress a modular reality into a dense parameter space, resulting in optimization conflicts. Then, we model a naive Routing-based Agentic AI and demonstrate that even a merely routing-based Agentic AI can beat the monolithic model exponentially in both sample and parameter complexity.

### 3.1. The Monolithic Dilemma

Let the parameter space be $\Theta \subseteq \mathbb{R}^d$. The goal of a monolithic model is to minimize the weighted average risk:

$$\theta^*_{\text{mono}} = \underset{\theta \in \Theta}{\arg\min} \, \mathcal{L}_{\text{total}}(\theta) = \underset{\theta \in \Theta}{\arg\min} \sum_{k=1}^{K} \alpha_k \mathcal{L}_k(\theta)$$

In contrast, a specialist model for task $k$ seeks the task-specific optimum $\theta^*_k = \arg\min_{\theta \in \Theta} \mathcal{L}_k(\theta)$.

**Assumption 3.1** (Regularity under Ideal Task Sharding).
Assuming the tasks are perfectly sharded such that each $\mathcal{D}_k$ represents a distinct, internally consistent function, the loss function $\mathcal{L}_k(\theta)$ is well-behaved. Specifically, we assume $\mathcal{L}_k(\theta)$ is twice continuously differentiable ($C^2$). Furthermore, in the local neighborhood of its optimal parameter $\theta^*_k$, $\mathcal{L}_k(\theta)$ is strictly convex, implying that its Hessian matrix $H_k(\theta) = \nabla^2 \mathcal{L}_k(\theta)$ is positive definite (PD), i.e., $v^\top H_k v > 0$ for all $v \neq 0$.

**Assumption 3.2** (Lipschitz Continuous Hessian). For each task $k$, the loss function $\mathcal{L}_k$ is twice differentiable and has a $\rho_k$-Lipschitz continuous Hessian, i.e., $\|\nabla^2 \mathcal{L}_k(\theta) - \nabla^2 \mathcal{L}_k(\theta')\| \leq \rho_k \|\theta - \theta'\|$.

We now state the proposition, which provides a lower bound on the monolithic risk. Rather than simple degradation, it demonstrates the inevitability of a suboptimal compromise: to accommodate the conflicting gradients of heterogeneous tasks, the monolithic model is forced to sacrifice peak proficiency in specialized domains, resulting in a flattened and averaged performance profile.

**Proposition 3.3** (The Average Trap). *Let $\mathcal{L}_{total}(\theta^*_{mono})$ be the converged risk of the monolithic model. Under Assumption 3.1, if the tasks are heterogeneous such that their optimal parameters do not coincide (i.e., $\exists i, j, \theta^*_i \neq \theta^*_j$), strictly positive lower bound $\epsilon > 0$ exists:*

$$\mathcal{L}_{\text{total}}(\theta^*_{\text{mono}}) \approx \sum_{k=1}^{K} \alpha_k \mathcal{L}_k(\theta^*_k) + \underbrace{\sum_{k=1}^{K} \frac{\alpha_k}{2} \|\theta^*_{\text{mono}} - \theta^*_k\|^2_{H_k}}_{\epsilon}$$

*where $\|v\|^2_{H_k} = v^\top H_k v$ denotes the squared Mahalanobis distance induced by the task curvature.*

See Appendix A.1 for the proof. Thus, we formally prove the inevitability of the "Generalist's Penalty": as the diversity of tasks increases, a monolithic model must trade away its expert-level acuity to maintain stability, resulting in a representation that is broadly usable but universally distinct from the optimum.

### 3.2. A Merely Routing-based Agentic AI Dominates

The limitations of the Monolithic learner, as proven in Theorem 3.3, stem from its attempt to approximate a global

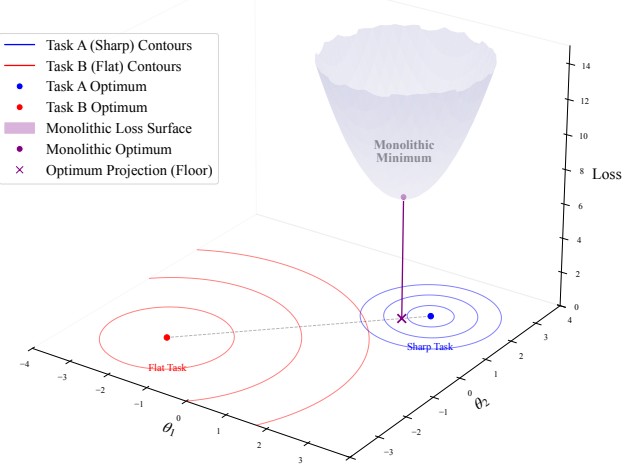

*Figure 2.* A demonstration of the Average Trap. The monolithic optimum is pulled towards the sharp task, illustrating the curvature-induced bias described in Proposition 3.3.

function over the complex union $\bigcup \mathcal{M}_k$. It is forced to smooth over the discontinuities between disjoint manifolds, expending capacity on the empty ambient space.

Now, we formalize a naive Routing-based Agentic AI (denoted as $M_{\text{R-Agentic}}$), which in contrast, bypasses the Average Trap by explicitly aligning its architecture with the topological structure of $\mathcal{D}_{\text{real}}$. Instead of solving for a compromised global optimum, the system exploits the geometric decomposability of the task mixture. We formalize the routed agentic hypothesis by assuming the target function $f$ can be factorized through a routing mechanism $\pi$ and a set of local maps:

$$f_{\text{R-Agentic}}(x) = \sum_{k=1}^{K} \mathbb{I}[\pi(x) = k] \cdot f_k(\phi_k(x))$$

where $\pi : \mathcal{X} \to \{1, \dots, K\}$ acts as a geometric router identifying the active manifold, and $\phi_k : \mathcal{M}_k \to \mathbb{R}^{d_k}$ represents the local coordinate chart (or projection) that maps the high-dimensional input onto the low-dimensional intrinsic manifold of task $k$.

In this analysis, we focus on the over-parameterized regime ($P \to \infty$), assuming the model possesses sufficient capacity to fully interpolate the finite training set. Under this assumption, the generalization error is no longer bottlenecked by model expressivity, but is strictly governed by the sample complexity relative to the intrinsic geometry of the data.

**Monolithic Baseline** Consider a Monolithic Learner $M_{\text{mono}}$ that attempts to approximate $f$ directly in the joint space $\mathbb{R}^D$. In the absence of structural assumptions, the model must populate the entire $D$-dimensional domain.

Given a fixed training budget of $N$ samples, the generalization error $\mathcal{E}_{\text{mono}}$ follows the standard convergence rate for Lipschitz functions in high-dimensional spaces by Proposition 2.2:

$$\mathcal{E}_{\text{mono}}(N) \approx \mathcal{O}\left(N^{-\frac{1}{D}}\right)$$

This relationship highlights that the error convergence is bottlenecked by the total dimension $D$. As the task complexity ($D$) increases linearly, the sample size required to maintain a constant error rate grows exponentially ($N \propto \epsilon^{-D}$).

**Routing-based Agentic Decomposition** In the Routing-based Agentic AI framework, the problem is explicitly decomposed into $K$ distinct sub-tasks. Each agent $A_k$ is responsible for learning a sub-function $f_k : \mathbb{R}^{d_k} \to \mathbb{R}$. Assuming the aggregation (or routing) function $\pi$ is fixed or introduces negligible error, the system's complexity is determined by the complexity of its sub-components.

Assuming the training budget $N$ is distributed among the agents (e.g., $N/K$ samples per agent), the total error bound is dominated by the sub-task with the highest dimensionality. Let $d_{\max} = \max_k(d_k)$ and let $L_k$ be the Lipschitz constant for $f_k$. The error upper bound for Routing-based Agentic AI is given by:

$$\mathcal{E}_{\text{R-Agentic}}(N) = \sum_{k=1}^{K} \mathcal{E}_k \approx \sum_{k=1}^{K} L_k \cdot \mathcal{O}\left(\left(\frac{N}{K}\right)^{-\frac{1}{d_k}}\right) \quad (2)$$

$$\leq \sum_{k=1}^{K} L_k \cdot \mathcal{O}\left(\left(\frac{N}{K}\right)^{-\frac{1}{d_k}}\right) + \mathcal{E}_{\text{routing}}$$

Assuming ideal routing and considering the dominance of the most complex sub-task, we obtain:

$$\mathcal{E}_{\text{R-Agentic}}(N) \approx \mathcal{O}\left(K \cdot N^{-\frac{1}{d_{\max}}}\right)$$

Since $d_{\max} \ll D$, the exponent $-1/d_{\max}$ is significantly larger in magnitude (more negative) than $-1/D$, implying a substantially faster decay of error.

We further quantify the advantage of the Routing-based Agentic AI by comparing the ratio of the expected errors. Neglecting constant factors, we derive the following relation:

$$\frac{\mathcal{E}_{\text{R-Agentic}}(N)}{\mathcal{E}_{\text{mono}}(N)} \approx \frac{K \cdot N^{-\frac{1}{d_{\max}}}}{N^{-\frac{1}{D}}} = K \cdot N^{\left(\frac{1}{D} - \frac{1}{d_{\max}}\right)}$$

Since $d_{\max} \ll D$, the exponent $\left(\frac{1}{D} - \frac{1}{d_{\max}}\right)$ is strictly negative, indicating that the error of Routing-based Agentic AI vanishes exponentially faster relative to the Monolithic error as $N$ grows.

The implication of the negative exponent is profound when interpreted through the lens of sample complexity. Specifically, to achieve a target error rate $\epsilon$, the Monolithic model

requires $N_{\text{mono}} \propto \epsilon^{-D}$ samples, whereas the Routing-based Agentic AI requires only $N_{\text{R-Agentic}} \propto K^{d_{\max}} \epsilon^{-d_{\max}}$. The ratio of data requirements is:

$$\frac{N_{\text{R-Agentic}}}{N_{\text{mono}}} \propto K^{d_{\max}} \epsilon^{D-d_{\max}} \quad (3)$$

Since $\epsilon$ is typically small ($\epsilon \ll 1$) and the dimensionality gap $D - d_{\max}$ is substantial, the term $\epsilon^{D-d_{\max}}$ asymptotically dominates the ratio. Although the pre-factor $K^{d_{\max}}$ introduces a polynomial overhead corresponding to the number of agents, it is negligible compared to the exponential reduction driven by the dimensionality reduction as $\epsilon \to 0$.

This aligns with empirical evidence that specialized agents are significantly more data-efficient. (Hu et al., 2022). By decomposing the high-dimensional manifold into lower-dimensional ones, the Routing-based Agentic AI effectively circumvents the Curse of Dimensionality, transforming an overwhelming learning problem into a set of solvable ones.

Based on the scaling law $\mathcal{E}(P) \propto P^{-\frac{\kappa}{d}}$ established in Eq. (1), parameter efficiency follows the same dimensionality-dependent power law as sample efficiency. Consequently, the analysis above applies symmetrically to model size: the Monolithic learner's error decay is stifled by the ambient dimension ($\mathcal{O}(P^{-\frac{\kappa}{D}})$), whereas the Routing-based Agentic AI benefits from a faster rate governed by the lower intrinsic dimension ($\mathcal{O}(P^{-\frac{\kappa}{d_{\max}}})$).

**The Routing Regret** We now analyze the omitted $\mathcal{E}_{\text{routing}}$ and explain why it can be ideally omitted in the inequality (2). We define the *Routing Regret*, denoted as $\mathcal{E}_{\text{routing}}$, as the expected performance deficit caused by selecting a sub-optimal expert. Formally, let $k^*(x)$ be the index of the optimal expert for input $x$, and $\pi(x)$ be the expert selected by the router. The routing error can be decomposed into the probability of error and the severity of the mismatch:

$$\mathcal{E}_{\text{routing}} = \mathbb{E}_{x \sim \mathcal{D}_{\text{real}}}\Big[\underbrace{\mathbb{I}(\pi(x) \neq k^*(x))}_{\epsilon_\pi : \text{Routing Error Rate}} \cdot \underbrace{\left(L(A_{\pi(x)}(x)) - L(A_{k^*(x)}(x))\right)}_{\Delta(x) : \text{Mismatch Penalty}}\Big]$$

To derive a tractable bound, we analyze the two components of this expectation: the Routing Error Rate ($\epsilon_\pi$) and the Mismatch Penalty ($\Delta$).

The router essentially solves a $K$-way classification problem, mapping the input space $\mathcal{X}$ to the set of agent indices $\mathcal{Y} = \{1, \ldots, K\}$. The hardness of this task is governed by the complexity of the hypothesis class $\mathcal{H}_{\text{router}}$ employed by the router.

We quantify the Routing Error Rate $\epsilon_\pi$ as the generalization error of the router. Invoking Theorems from Section 2.3, for the most common routers trained on $N_{\text{router}}$ samples, we further obtain how the bounds scale with the number of

agents $K$ as follows:

$$\epsilon_\pi \propto \begin{cases} \tilde{\mathcal{O}}\left(\frac{\log K}{\sqrt{N_{\text{router}}}}\right), & \text{if } \pi \text{ is a Tree-based Router} \\ \sqrt{\frac{K}{N_{\text{router}}}}, & \text{if } \pi \text{ is a Neural Router} \end{cases}$$

The routing error rate of both kinds of routers increases as $K$ increases, though for the tree-based one, the polylogarithmic dependence allows a stronger guarantee for the scalability.

The severity of a routing error depends on the orthogonality of the experts. We define the maximum mismatch penalty as $\Delta_{\text{max}} = \sup_{x, j \neq k^*} |L(A_j(x)) - L(A_{k^*}(x))|$.

We verify the intuition that the cost of mismatch $\Delta_{\text{max}}$ scales with the granularity of specialization $K$. We model this relationship using subspace information loss.

**Assumption 3.4** (Manifold Alignment with Orthogonal Subspaces). Following Definition 2.1, we assume each manifold $\mathcal{M}_k$ is contained within a feature subspace $S_k \subset \mathbb{R}^D$. These subspaces form an orthogonal decomposition of the feature space, such that $\mathbb{R}^D = \bigoplus_{k=1}^K S_k$ and $S_j \perp S_k$ for $j \neq k$.

**Lemma 3.5.** *Let $x \sim \mathcal{D}_j$ be an input belonging to task $j$. Ideally, the information required to solve $x$ is contained in $S_j$. However, if misrouted to expert $A_k$ ($k \neq j$), the expert processes the projection $P_k x$. The information preservation ratio $\rho$ is given by the cosine similarity between the required subspace and the expert's subspace:*

$$\rho(j, k) = \frac{\|P_k x\|^2}{\|x\|^2}$$

*Under Assumption 3.4, if $j \neq k$, then $S_j \perp S_k$, implying $P_k x \approx 0$. In a relaxed setting with partial overlap, as $K$ increases, the subspaces become increasingly disjoint. We model the residual information as inversely proportional to $K$:*

$$\mathbb{E}[\|P_k x\|^2] \propto \frac{1}{K-1}\|x\|^2 \quad (\text{for } j \neq k)$$

Let the loss function $L$ be $\lambda$-Lipschitz continuous. The mismatch penalty is bounded by the distance in the feature space caused by the projection loss:

$$\Delta(x) \leq \lambda \|x - P_k x\| = \lambda \|x\| \left(1 - \frac{\|P_k x\|}{\|x\|}\right)$$

Substituting the expected information preservation ratio from Lemma 3.5, and defining the maximum potential loss on the domain as $L_{\text{max}} \triangleq \lambda \mathbb{E}[\|x\|]$ (representing the loss scaling with input magnitude), we obtain:

$$\Delta_{\text{max}}(K) \approx L_{\text{max}} \left(1 - \sqrt{\frac{1}{K-1}}\right) \sim L_{\text{max}} \left(1 - \frac{1}{\sqrt{K}}\right)$$

asymptotically, as $K \to \infty$, using the Taylor expansion $(1 - \epsilon)^{-1/2} \approx 1 + \epsilon/2$.

Finally, combining the routing error rate and the mismatch penalty, we derive an upper bound for the Routing Regret:

$$\mathcal{E}_{\text{routing}} \leq \begin{cases} C_{\text{tree}} L_{\text{max}} \left(1 - \frac{1}{\sqrt{K}}\right) \\ \quad \times \sqrt{\frac{\text{poly}(\log K)}{N_{\text{router}}}}, & \text{if Tree-based Router} \\ C_{\text{NN}} L_{\text{max}} \left(1 - \frac{1}{\sqrt{K}}\right) \\ \quad \times \sqrt{\frac{K}{N_{\text{router}}}}, & \text{if Neural Router} \end{cases}$$

**Joint Bound and Optimal Granularity** The preceding analysis factored out routing error for clarity. We now present a joint bound that unifies specialization gain and routing cost. Substituting the routing error $\epsilon_\pi$ and mismatch $\Delta$ into the agentic bound (2):

$$\mathcal{E}_{\text{R-Agentic}}(K, N) \leq \underbrace{\frac{K C_{\text{exp}}}{N^{1/d_{\text{max}}}}}_{\text{decreases with } K} + \underbrace{\Delta_{\text{max}}(K) \cdot \epsilon_\pi(K)}_{\text{increases with } K}$$

This yields a U-shaped error profile in $K$: too few agents ($K \to 1$) provides insufficient specialization, while too many ($K \to \infty$) causes routing overhead to dominate, with an optimal $K^*$ in between. Expanding for specific router types:

For tree-based routers:

$$\mathcal{E} \leq \frac{KC}{N^{1/d_{\text{max}}}} + C_{\text{tree}} L_{\text{max}} \left(1 - \frac{1}{\sqrt{K}}\right) \sqrt{\frac{\text{poly}(\log K)}{N_{\text{router}}}}$$

The modularity cost grows polylogarithmically, so specialization dominates for large $K$.

For neural routers:

$$\mathcal{E} \leq \frac{KC}{N^{1/d_{\text{max}}}} + C_{\text{NN}} L_{\text{max}} \left(1 - \frac{1}{\sqrt{K}}\right) \sqrt{\frac{K}{N_{\text{router}}}}$$

The cost rises as $\sqrt{K}$, restricting $K^*$ unless $N_{\text{router}} \propto K$. In both cases, the specialization gain ($N^{-1/d_{\text{max}}}$ vs. $N^{-1/D}$) dominates the polynomial routing cost for sufficiently large $N$, since $d_{\text{max}} \ll D$.

Consequently, for a fixed data budget, the optimal number of agents $K^*$ is the solution to $\frac{\partial \mathcal{E}_{\text{total}}}{\partial K} = 0$. System designers face a dichotomy: use tree-based routing to maximize scalability ($K \gg 1$) or neural routing to handle complex, non-axis-aligned task boundaries at the cost of a smaller feasible agent pool.

In a data-scarce regime, the tree-based router is superior. Its error scales with $\mathcal{O}(\sqrt{\log K})$, allowing massive scaling of $K$ even with limited routing data. Here, the Routing Regret is negligible. In a data-rich regime, if the sample size $N$ is sufficient ($N \gg K$), the linear penalty of neural routers ($\mathcal{O}(\sqrt{K})$) is suppressed by the large denominator.

In this regime, neural routing becomes preferable despite its higher sample complexity, as it avoids the inductive bias of trees and can capture complex, non-hierarchical expert boundaries.

Now we establish that the transition from Monolithic to Routing-based Agentic AI is not merely an architectural preference, but a geometric necessity for mastering heterogeneous, high-dimensional tasks, manifested by real-world task distribution.

## 4. A Closer Look at General Agentic AI

Through the analysis in Section 3, we have theoretically established that decomposing a monolithic problem into specialized sub-tasks aligns with the real-world task distribution and yields exponential gains in efficiency and effectiveness. However, it primarily modeled the system as a static routing between expert agents. In real-world Agentic AI, agents rarely operate in isolation; they function as interconnected nodes facilitating the dynamic propagation of information.

To rigorously analyze the generalization bounds, we first establish a formal mathematical definition of Agentic AI. Unlike monolithic models, which approximate a target function $F : \mathcal{X} \rightarrow \mathcal{Y}$ via a single dense parameterization, Agentic AI is defined as a structured composition of specialized operators.

**Definition 4.1** (Agentic AI as a System of a Topological Compositional DAG of Agents). Let $\mathcal{X}$ be the global input space and $\mathcal{Y}$ be the global output space. An Agentic AI system is defined as a tuple $\Psi = (\mathcal{G}, \mathcal{F}, \Lambda)$, where:

1. $\mathcal{G} = (\mathcal{V}, \mathcal{E})$ is a Directed Acyclic Graph (DAG) with $K = |\mathcal{V}|$ nodes, representing the flow of information. The node set $\mathcal{V}$ is topologically sorted.

2. $\mathcal{F} = \{f_1, \ldots, f_K\}$ is a set of heterogeneous, learnable mappings (agents). Each agent $v_i$ implements a local function $f_i : \mathcal{H}_{in}^{(i)} \times \Theta_i \rightarrow \mathcal{H}_{out}^{(i)}$, where $\Theta_i$ represents the agent's specific parameters and $\mathcal{H}$ represents the latent manifold of intermediate representations.

3. $\Lambda$ is a composition operator that maps the outputs of parent nodes to the input of a child node. For any agent $v_i$, the input state $s_i$ is constructed from the set of parents $Pa(i) = \{v_j \mid (v_j, v_i) \in \mathcal{E}\}$:

$$x_i = f_i \left( \Lambda \left( \{x_j\}_{j \in Pa(i)} \right) ; \theta_i \right)$$

The global system behavior is not a static function, but rather emerges as the topological flow from the source nodes (initialized by $\mathcal{X}$) to the sink nodes (projected to $\mathcal{Y}$), respecting the partial order of $\mathcal{G}$.

For each node $v_i$, let $S_i \in \{0, 1\}$ be a Bernoulli random variable indicating the success of the specific task assigned to agent $i$. The execution of $v_i$ depends on the latent states or outputs $h_{Pa(i)}$ from its parent set $Pa(i) = \{v_j \mid (v_j, v_i) \in \mathcal{E}\}$. Assuming the Markov property on the graph, the joint probability of a successful execution trajectory is:

$$P(S_1, \ldots, S_N) = \prod_{i=1}^{N} P(S_i \mid h_{Pa(i)})$$

Then, we transform the multiplicative success probability into an additive loss function using the negative log-likelihood. The loss for Agentic AI $\mathcal{L}_{\text{Agentic}}$ can be defined as:

$$\mathcal{L}_{\text{Agentic}}(\boldsymbol{\theta}) = -\log \left( \prod_{i=1}^{K} P(S_i = 1 \mid h_{Pa(i)}) \right)$$

$$= \sum_{i=1}^{K} \underbrace{-\log P(S_i = 1 \mid h_{Pa(i)})}_{\ell_i(\theta_i)}$$

where $\ell_i$ represents the local loss contribution of agent $i$.

To explicitly derive the local loss $l_i$, we instantiate the abstract local function $f_i$ as a stochastic generator parameterized by a policy. Specifically, the execution of $f_i$ corresponds to sampling an action $a_i$ (which constitutes the output $x_i$) from a policy $\pi_{\theta_i}(a_i \mid s_i)$ conditioned on the input state $s_i$. Consequently, the local loss $\ell_i$ relates to the agent's policy via the expectation over actions:

$$\ell_i(\theta_i, s_i) = -\log \left( \int_{\mathcal{A}} \rho(s_i, a_i) \pi_{\theta_i}(a_i \mid s_i) \, da_i \right)$$

where $\rho(s_i, a_i) \in [0, 1]$ is the conditional success probability of taking action $a_i$ in state $s_i$.

To understand how the Agentic AI generalizes, we must quantify how a local perturbation at a specific agent propagates through complex topologies to affect the loss.

We define the *Direct Adjacency Jacobian Matrix* $\mathbf{J} \in \mathbb{R}^{K \times K}$. The entry $J_{ji}$ captures the local sensitivity of agent $j$ to its direct parent agent $i$:

$$J_{ji} = \frac{\partial x_j}{\partial x_i} = \begin{cases} \frac{\partial f_j}{\partial x_i} & \text{if } (i,j) \in \mathcal{E} \\ 0 & \text{otherwise} \end{cases}$$

Then, we can derive the topological weight of a specific agent in the DAG.

**Lemma 4.2** (Topological Weight). *Let $\mathcal{L}$ be the Agentic AI loss function and $\omega_u = \left\| \frac{d\mathcal{L}}{dx_u} \right\|$ be the scalar Topological Weight representing the total sensitivity of the loss to agent $u$. The weight $\omega_u$ is determined by the aggregation of gradient flow along all paths connecting $u$ to the sink agents:*

$$\omega_u = \left\| \sum_{v \in Sinks} \frac{\partial \mathcal{L}}{\partial x_v} \sum_{\rho \in Paths(u \rightarrow v)} \left( \prod_{(a,b) \in \rho} J_{ba} \right) \right\|$$

See Appendix A.2 for the proof. Given specific agent weights, we analyze the Agentic AI generalization error, $\mathcal{E}_{\text{Agentic}}$. Consistent with Section 3, we assume local errors decay via a power law governed by intrinsic dimension $d_u$. Using a first-order Taylor expansion around the optimal agent outputs, $\mathcal{E}_{\text{Agentic}}$ is approximated as the weighted superposition of local errors.

$$\mathcal{E}_{\text{Agentic}} \approx \sum_{u=1}^{K} \omega_u \cdot \mathcal{E}_u \approx \sum_{u=1}^{K} \omega_u \cdot \mathcal{O}\left( \left( \frac{N}{K} \right)^{-\frac{1}{d_u}} \right) \quad (4)$$
$$\approx C(G) \cdot \left( \frac{N}{K} \right)^{-\frac{1}{d_{\text{eff}}}}$$

where $d_{\text{eff}}$ is the effective intrinsic dimension of the task and $C(G)$ is the Topology Factor determined by the topology of Agentic AI.

To disentangle the impact of the Topology Factor from the intrinsic difficulty of specific sub-tasks, we assume that the complex global task is divided into sub-tasks of comparable intrinsic difficulty, formally for any sub-task $u$, $d_u \approx d_{\text{eff}}$. Then, the convergence rate term becomes uniform across all agents. This allows us to factor the complexity term out of the summation in Equation (4), isolating the Topology Factor. And further, the Topology Factor $C(G)$ can be formally defined as the sum of Topological Weights:

$$C(G) \equiv \sum_{u=1}^{K} \omega_u = \sum_{u=1}^{K} \left\| \sum_{v \in \text{Sinks}} \frac{\partial \mathcal{L}}{\partial x_v} \left( \sum_{\rho \in \text{Path}_{u \to v}} \prod_{e \in \rho} J_e \right) \right\|$$

The definition allows us to analyze the stability of different agentic orchestrations by evaluating how $C(G)$ scales with DAG complexity, and confirms that while $d_{\text{eff}}$ governs the rate of convergence, $C(G)$ determines the magnitude of the error. Agentic AI succeeds when the topology minimizes $C(G)$ while maximizing the dimensionality gap.

**Theorem 4.3** (Agentic AI Convergence Superiority). *As the scale of resources (dataset size $N$ or parameter budget $P$) increases, the generalization error of the Agentic AI decays exponentially faster than that of the Monolithic model, provided the topology satisfies spectral stability (well designed with $C(G) < \infty$).*

Apart from the overall analysis of the graph, we can further decompose global instability into single-edge contributions to better analyze the connections.

**Lemma 4.4** (Topological Edge Weight). *Consider a specific edge $e^* = (u, v)$ connecting a parent agent $u$ to a child agent $v$. The Topological Edge Weight $\mathcal{W}(e^*)$ represents the total gradient flux passing through this edge, linking the accumulated history of the parent to the future criticality of*

*the child. It is formally defined as:*

$$\mathcal{W}(e^*) = \underbrace{\left( 1 + \sum_{k \in \mathcal{P}(u)} \left\| \sum_{\rho \in \text{Path}(k \to u)} \prod_{e \in \rho} J_e \right\| \right)}_{\text{Upstream History}}$$
$$\cdot \underbrace{\| J_{e^*} \|}_{\text{Local Valve}} \cdot \underbrace{\left\| \sum_{z \in \text{Sinks}} \frac{\partial \mathcal{L}}{\partial x_z} \sum_{\gamma \in \text{Path}(v \to z)} \prod_{e' \in \gamma} J_{e'} \right\|}_{\text{Downstream Future}}$$

*where $\mathcal{P}(u)$ is the set of predecessors of agent $u$, and* Sinks *denotes the set of final output agents.*

See Appendix A.3 for the proof. To minimize the global error $C(G)$, we must well orchestrate the necessary agents and find the small combinations of edge weights. This equation reveals a fundamental design principle regarding what kind of edges we should build *at runtime* and gives a reference to analyze each edge *post hoc*. Specifically: (1) after long chains (high Upstream History), edges must be contractive ($\| J_{e^*} \| < 1$) to filter accumulated noise, e.g., critic or judge edges. (2) Before critical decisions (high Downstream Sensitivity), edges should satisfy $\| J_{e^*} \| \ll 1$, e.g., voting or verification edges that collapse multiple paths into a stable signal.

Consequently, we find that optimal edges function as adaptive valves. An ideal edge is not a passive pipe but an active variational filter that suppresses the noise accumulated from the upstream before it propagates to critical downstream tasks.

Finally, in this section, we not only extend the superiority from a Routing-based Agentic AI to a general Agentic AI by deriving the topological properties of the Agentic AI, but also further analyze the impact of specific agents and edges, giving backgrounds for Agentic AI design and Agentic AI success and failure explanation.

## 5. Alternative Views

**Monolithic scaling is enough for AGI** DeepMind ever stated that *"...an agent capable on a large number of tasks and able to be adapted with little extra data to succeed at an even larger number of tasks can be obtained by scaling data, compute and parameters..."* (Reed et al., 2022). Agüera y Arcas & Norvig (2023) even said that *"...the most important parts of it (AGI) have already been achieved by the current generation of advanced AI large language models such as ChatGPT, Bard, LLaMA and Claude."* However, very few researchers firmly admit that AGI has come and are one hundred percent satisfied with one specific monolithic model, at least in coding, not mention in all real-world tasks. We don't object to the view that scaling is effective, but both neural scaling laws and empirical experiments

have demonstrated that the marginal improvement is more obvious as an inevitable bottleneck (Kaplan et al., 2020; Hoffmann et al., 2022). More attention should be drawn to Agentic AI to break the bottleneck.

**Agentic AI is conceptually similar to Mixture-of-Experts** Mixture-of-Experts (MoE) (Shazeer et al., 2017) and Agentic AI share a common design principle: routing inputs to specialized sub-networks rather than processing everything monolithically. The empirical success of sparse MoEs (Fedus et al., 2022; Lepikhin et al., 2021) validates this core premise of our theory. In our framework, MoE corresponds to the routing regime of Section 3.2, where $C(G) \approx \sum L_u$ and the system is inherently stable.

However, Agentic AI generalizes beyond MoE in three fundamental aspects. First, in scope: MoE employs fixed expert sub-networks with learned gating within a single forward pass, whereas Agentic AI deploys autonomous agents with independent parameters capable of multi-step reasoning (Sapkota et al., 2026). Second, in topology: MoE implements single-layer routing (router $\rightarrow$ expert), while Agentic AI extends to arbitrary DAG compositions as formalized in Section 4. Third, in routing mechanism: MoE relies on differentiable gating trained end-to-end, whereas agentic routing accommodates iterative refinement, external tool use, and dynamic knowledge retrieval (Sapkota et al., 2026; Anthropic, 2025). While MoE and routing-based Agentic AI share a common design principle, Agentic AI extends to richer topological structures with greater expressivity.

**Multi-Agent systems often fail** Empirical evidence indicates that increasing agent quantity often introduces organizational entropy rather than performance gains. Complexity frequently hinders reliability, with failures stemming primarily from system design issues, inter-agent misalignment, and task verification difficulties (Pan et al., 2025). Furthermore, current LLMs struggle with coordination tasks requiring Theory of Mind compared to RL methods (Agashe et al., 2025), necessitating dedicated automated methods to diagnose these persistent failures (Zhang et al., 2025a).

Recent works exploring LaMAS flaws align with our derivation, attributing failures to Topological Weights and Edge Weights. For instance, misaligned agents introduce toxic topological properties, causing massive downstream variance and hallucination. This necessity for topological awareness is exemplified by the performance surge with well-designed topologies (Anthropic, 2025). Agentic AI demands dedicated topological design; most current frameworks are merely static pipeline decompositions based on human priors, masquerading as true Agentic AI.

## 6. Call to Action

**Prioritize Agentic AI for accessible AGI research** We urge researchers and institutions, especially those with limited resources, to prioritize Agentic AI, which offers a viable alternative to the prohibitive costs of monolithic scaling and yields exponential gains in both sample and parameter efficiency. This paradigm allows for state-of-the-art generalization without the need for brute-force computation. Since the efficiency advantage grows exponentially with the dimensionality gap between the ambient space and task-intrinsic manifolds, a well-designed agentic system of moderately sized specialists can match or exceed monolithic performance at a fraction of the cost, broadening access to AGI research beyond resource-rich laboratories.

**Not only fine-tune weights, but also invent better multi-agent evolution methods for applicable Agentic AI** The community must expand its focus from simply fine-tuning individual agents to a broader and more diverse optimization of the agentic system. Research should look beyond specific weight adjustments and explore various enhancements, such as mitigating organizational entropy, designing graph, tree or forest evolution methods, and ensuring spectral stability. The goal is to move from static pipelines to the evolution of topologically stable multi-agent ecosystems. In particular, automated methods for discovering optimal DAG topologies, routing mechanisms that scale gracefully with agent count, and topology-aware evaluation protocols that attribute failures to specific graph components are all pressing open problems.

## 7. Conclusion

This paper challenges the dogma of monolithic scaling, identifying Agentic AI as the superior pathway to AGI. By formalizing real-world task distributions as unions of low-dimensional manifolds, we prove that monolithic models suffer an irreducible Average Trap penalty that accumulates with task diversity. In contrast, even routing-based Agentic AI achieves exponentially superior sample and parameter efficiency, as each agent operates on a low-dimensional sub-manifold ($d_k \ll D$) rather than the full ambient space. We extend this to general DAG topologies, introducing Compositional Capacity $C(G)$ and Edge Weight decomposition $\mathcal{W}(e^*)$ for principled multi-agent design, and show that the agentic advantage degrades gracefully under partial task overlap with an optimal granularity $K^*$ balancing specialization against routing costs. We further clarify the relationship to Mixture-of-Experts and attribute current multi-agent failures to poor topological design rather than fundamental flaws. Ultimately, we conclude that achieving AGI requires shifting from brute-force scaling to the precise optimization of stable, well-designed Agentic AI ecosystems.

# Acknowledgements

This work was supported by National Natural Science Foundation of China (62322603) and Shanghai Municipal Special Program for Basic Research on General AI Foundation Models (Grant No. 2025SHZDZX025D08).

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

# A. Proofs for Theorems

## A.1. Proof of Proposition 3.3

*Proof.* Since $\theta^*_{\text{mono}}$ is a minimizer of the total convex loss $\mathcal{L}_{\text{total}}$, it satisfies the first-order optimality condition:

$$\nabla\mathcal{L}_{\text{total}}(\theta^*_{\text{mono}}) = \sum_{k=1}^{K} \alpha_k \nabla\mathcal{L}_k(\theta^*_{\text{mono}}) = 0$$

This implies that the weighted gradients sum to zero. Unless all $\theta^*_k$ are identical, for any specific task $k$, $\nabla\mathcal{L}_k(\theta^*_{\text{mono}}) \neq 0$. Geometrically, $\theta^*_{\text{mono}}$ lies within the convex hull of the individual optima $\{\theta^*_k\}_{k=1}^{K}$ but coincides with none. Thus, the monolithic solution is a compromise, merely a Pareto stationary point where gradients cancel out destructively.

We expand the loss $\mathcal{L}_k(\theta)$ for each task around its specific optimum $\theta^*_k$. By Assumption 3.1, $\nabla\mathcal{L}_k(\theta^*_k) = 0$. Using the Lagrangian form of the Taylor expansion, we have:

$$\mathcal{L}_k(\theta^*_{\text{mono}}) = \mathcal{L}_k(\theta^*_k) + \frac{1}{2}(\theta^*_{\text{mono}} - \theta^*_k)^\top H_k(\theta^*_{\text{mono}} - \theta^*_k) + R_3(\theta^*_{\text{mono}}, \theta^*_k)$$

where $R_3$ is the third-order remainder term. Under the assumption of $\rho$-Lipschitz Hessian (Assumption 3.2), this remainder is bounded by $|R_3| \leq \frac{\rho}{6}\|\theta^*_{\text{mono}} - \theta^*_k\|^3$.

Substituting this back into the total risk objective:

$$\mathcal{L}_{\text{total}}(\theta^*_{\text{mono}}) = \sum_{k=1}^{K} \alpha_k \left[\mathcal{L}_k(\theta^*_k) + \frac{1}{2}\|\theta^*_{\text{mono}} - \theta^*_k\|^2_{H_k} + R_3^{(k)}\right]$$

$$= \underbrace{\sum_{k=1}^{K} \alpha_k \mathcal{L}_k(\theta^*_k)}_{\mathcal{L}_{\text{ideal}}} + \sum_{k=1}^{K} \frac{\alpha_k}{2}\|\theta^*_{\text{mono}} - \theta^*_k\|^2_{H_k} + \underbrace{\sum_{k=1}^{K} \alpha_k |R_3^{(k)}|}_{\text{Higher-order error}}$$

To guarantee that the interference cost $\epsilon$ is significant, the quadratic term must dominate the third-order error. Since $\|\cdot\|^2_{H_k}$ scales with $\Delta^2$ while the error scales with $\Delta^3$, for a local neighborhood around the optima where the task divergence is bounded, the positive curvature (guaranteed by positive definite $H_k$) strictly dominates the higher-order variations. Thus, we derive the lower bound:

$$\mathcal{L}_{\text{total}}(\theta^*_{\text{mono}}) \gtrsim \mathcal{L}_{\text{ideal}} + \sum_{k=1}^{K} \frac{\alpha_k}{2}\|\theta^*_{\text{mono}} - \theta^*_k\|^2_{H_k}$$

This confirms that $\epsilon > 0$ holds as long as the conflicting gradients force $\theta^*_{\text{mono}}$ away from individual optima, creating an irreducible quadratic penalty. $\qquad\square$

## A.2. Proof of Lemma 4.2

*Proof.* Since agents are topologically sorted, $\mathbf{J}$ is strictly lower triangular and nilpotent.

By the multivariate chain rule, the total variation $\frac{dx_j}{dx_i}$ captures the cumulative effect of agent $i$ on agent $j$ through *all* possible paths.

$$\frac{dx_j}{dx_i} = \underbrace{\frac{\partial x_j}{\partial x_i}}_{\text{Direct edge}} + \sum_{k\in Pa(j), k\neq i} \underbrace{\frac{\partial x_j}{\partial x_k}}_{\text{Direct path}} \cdot \underbrace{\frac{dx_k}{dx_i}}_{\text{Recursive path}} = \sum_k \underbrace{\frac{\partial x_j}{\partial x_k}}_{J_{jk}} \underbrace{\frac{dx_k}{dx_i}}_{M_{ki}}$$

Then we define $\mathbf{M} \in \mathbb{R}^{K\times K}$ be the *Influence Matrix* where $M_{ji} = \frac{dx_j}{dx_i}$. The recursive relation can be written in matrix form:

$$\mathbf{M} = \mathbf{J}\mathbf{M} + \mathbf{I}$$

Here, $\mathbf{I}$ is defined as the self-influence $\mathbf{I}_{ii} = \frac{dx_i}{dx_i}$ and anywhere else $\mathbf{0}$. Then, rearranging for $\mathbf{M}$:

$$(\mathbf{I} - \mathbf{J})\mathbf{M} = \mathbf{I} \implies \mathbf{M} = (\mathbf{I} - \mathbf{J})^{-1}$$

Since $\mathbf{J}$ is nilpotent (due to the acyclic property of DAGs), this inverse can be expanded as a Neumann Series:

$$\mathbf{M} = \sum_{k=0}^{K-1} \mathbf{J}^k = \mathbf{I} + \mathbf{J} + \mathbf{J}^2 + \dots$$

Physically, $\mathbf{J}^k$ represents the influence propagation along paths of length exactly $k$. The matrix $\mathbf{M}$ mathematically aggregates all parallel and serial paths automatically. Then, we can derive the topological weight of a specific agent in the DAG.

Let $\mathbf{g} = [\|\frac{\partial \mathcal{L}}{\partial x_1}\|, \dots, \|\frac{\partial \mathcal{L}}{\partial x_K}\|]^T$ be the gradient of the loss with respect to agent outputs (typically non-zero only for sink agents). The total sensitivity of $\mathcal{L}$ to a specific agent $u$ is the $u$-th component of:

$$\boldsymbol{\omega} = \mathbf{M}^T \mathbf{g}$$

The scalar Topological Weight $\omega_u$ for agent $u$ is explicitly:

$$\omega_u = \left\| \frac{d\mathcal{L}}{dx_u} \right\| = \left\| \sum_{v \in \text{Sinks}} \frac{\partial \mathcal{L}}{\partial x_v} \sum_{\rho \in \text{Paths}(u \to v)} \underbrace{\left( \prod_{(a,b) \in \rho} J_{ba} \right)}_{\text{Weight of path } \rho} \right\|$$

$\square$

### A.3. Proof of Lemma 4.4

*Proof.* The weight is derived by tracing the full back-propagation path of the loss gradient through the specific edge $e^*$. The total influence is the product of the signal magnitude reaching the parent $u$ and the distribution of that signal to all upstream ancestors.

First, we isolate the incoming gradient signal from the child $v$. By the chain rule, the gradient at $u$ contributed strictly by $v$ is $\frac{\partial \mathcal{L}}{\partial x_v} J_{e^*}$. The magnitude of this local flux is:

$$\|\text{Flux}_{v \to u}\| \leq \|J_{e^*}\| \cdot \left\| \frac{\partial \mathcal{L}}{\partial x_v} \right\| = \|J_{e^*}\| \cdot \omega_v$$

Second, we account for the propagation of this flux to the past. The signal distributes to $u$ itself (identity gain) and to every predecessor $k \in \mathcal{P}(u)$. The cumulative amplification factor is the sum of path gains:

$$\text{Upstream Mass} = \|I\| + \sum_{k \in \mathcal{P}(u)} \left\| \sum_{\rho \in \text{Path}(k \to u)} \prod_{e \in \rho} J_e \right\|$$

Third, we expand the downstream sensitivity $\omega_v$. The total gradient $\frac{\partial \mathcal{L}}{\partial x_v}$ is the aggregation of error signals back-propagated from all reachable sink nodes $z$. Expanding the influence matrix $M_{zv} = \frac{dx_z}{dx_v}$ as a sum over all paths $\gamma$:

$$\omega_v = \left\| \sum_{z \in \text{Sinks}} \frac{\partial \mathcal{L}}{\partial x_z} \frac{dx_z}{dx_v} \right\| = \left\| \sum_{z \in \text{Sinks}} \frac{\partial \mathcal{L}}{\partial x_z} \sum_{\gamma \in \text{Path}(v \to z)} \prod_{e' \in \gamma} J_{e'} \right\|$$

Multiplying these three components, Upstream Mass, Local Valve ($J_{e^*}$), and expanded Downstream Sensitivity, yields the complete Topological Edge Weight definition. $\square$

## B. Some Relevant Remarks

### B.1. Remark for Section 3.1

*Remark* B.1 (Data-Abundant Regime). It is crucial to distinguish our theoretical setup from typical few-shot transfer learning or low-resource multi-task learning scenarios. We assume a data-abundant regime distinct from low-resource settings, allowing models to reach convergence.

*Remark* B.2 (Alignment with Negative Transfer in Multi-Task Learning). The proposition of the Average Trap is consistent with the Negative Transfer in Multi-Task Learning, which states that when the gradient direction of task A makes an angle greater than 90 degrees with the gradient direction of task B, updating parameters by one step will simultaneously impair the performance of one or both tasks (Wang, 2021). Though researchers have been finding ways to mitigate the effect (Yu et al., 2020; Zhang et al., 2024), the $\epsilon$ term in the Average Trap is an unavoidable constraint.

*Remark* B.3 (When Is the Average Trap Binding?). The magnitude of the Average Trap penalty $\epsilon = \sum_{k=1}^{K} \frac{\alpha_k}{2} \|\theta_{\text{mono}}^* - \theta_k^*\|_{H_k}^2$ depends on both the number of tasks $K$ and the curvature $H_k$. For narrow task families (small $K$, closely related tasks), the penalty is mild: the task-optimal parameters $\theta_k^*$ cluster tightly, and the monolithic compromise remains near each individual optimum. However, the penalty becomes increasingly binding as the task distribution broadens toward AGI-level generality. First, $\epsilon$ accumulates with $K$ as the monolithic optimum must compromise across more divergent directions. Second, the effective ambient dimension $D$ of the union manifold $\bigcup \mathcal{M}_k$ grows with $K$, exacerbating the curse of dimensionality captured by Equation (3). This explains why monolithic models succeed on narrow benchmarks but progressively degrade on diverse leaderboards: the Average Trap transitions from a negligible correction to a dominant constraint precisely in the regime this paper targets.

## B.2. Remark for Assumption 3.1

*Remark* B.4 (Non-convexity of Neural Networks). As is well known, the loss surface of deep neural networks is non-convex, filled with saddle points and flat minima (Dauphin et al., 2014; Keskar et al., 2017). Even against a non-convex background, the conclusion still holds as long as conflicting gradient components exist (Liu et al., 2021).

## B.3. Remark for Section 3.2

*Remark* B.5 (LLMs as Non-parametric Estimators). While LLMs are parameterized by a finite set of weights, they operate in the "over-parameterized" regime where they function as universal approximators. In the context of complex task solving, we treat the LLM as a non-parametric estimator $\hat{f}_N$ attempting to learn the target function $\boldsymbol{f} : \mathbb{R}^D \to \mathbb{R}$ solely from data. Thus, the minimax lower bounds regarding sample complexity and dimensionality directly apply to the generalization capabilities of LLMs.

*Remark* B.6 (Data Abundance via Interaction). It is worth noting that, in the context of Agentic AI, the assumption of limited $N_{\text{router}}$ is often relaxed. Unlike static datasets, agents can continuously interact with the environment (the real world) to generate feedback. This online data generation capability implies that $N_{\text{router}}$ can be effectively infinite ($N \to \infty$), mitigating the sample complexity of neural routers and allowing the system to leverage their superior expressivity.

## B.4. Remark for Assumption 3.4

*Remark* B.7 (Graceful Degradation under Partial Overlap). Assumption 3.4 posits near-orthogonal task subspaces, which serves as a best-case analysis establishing the ceiling of the agentic advantage. In practice, tasks often share partial underlying structure. We formalize this relaxation by introducing an overlap parameter $\gamma = \dim(S_{\text{shared}})/d_{\max} \in [0, 1]$.

Each task's feature subspace decomposes as $S_k = S_{\text{shared}} \oplus S_k^{\text{res}}$, where $S_{\text{shared}}$ is the common subspace across all tasks with $\dim(S_{\text{shared}}) = \gamma \cdot d_{\max}$, and $S_k^{\text{res}}$ captures task-specific residuals with $S_j^{\text{res}} \perp S_k^{\text{res}}$ for $j \neq k$.

In the shared subspace, all tasks agree on a common optimum (zero penalty). In the residual subspace, the full divergence applies. The penalty from Proposition 3.3 therefore becomes:

$$\epsilon(\gamma) \approx (1 - \gamma) \cdot \epsilon_{\text{full}}$$

where $\epsilon_{\text{full}}$ is the penalty under full divergence ($\gamma = 0$).

The sample complexity ratio (Equation (3)) generalizes to:

$$\frac{N_{\text{R-Agentic}}}{N_{\text{mono}}} \propto K^{d_{\max}} \cdot \epsilon^{(1-\gamma)(D-d_{\max})}$$

The exponent shrinks linearly with $\gamma$ but remains strictly negative for any $\gamma < 1$, preserving the exponential advantage.

The mismatch penalty (Lemma 3.5) generalizes to:

$$\Delta_{\max}(K, \gamma) \approx L_{\max} \cdot (1 - \gamma) \cdot \left(1 - \frac{1}{\sqrt{K}}\right)$$

At $\gamma = 0$ (full separation), the original results are recovered exactly. At $\gamma > 0$, the advantage degrades smoothly but remains strictly positive for all $\gamma < 1$. At $\gamma = 1$ (complete overlap), the system reduces to the monolithic case, confirming that agentic decomposition provides a strict generalization of monolithic learning.

