# OpenReview forum: "Position: Agentic AI System Is a Foreseeable Pathway to AGI"
_ICML.cc/2026/Position_Paper_Track — ICML 2026 Position Paper Track regular_

### Official Review · Reviewer_VfDj · 2026-03-12

**Significance:** 3
**Argument Clarity:** 4
**Rating:** 5
**Confidence:** 4

**Questions:**

**Manifold Assumption**: Your analysis assumes each task lies on a distinct low-dimensional manifold Mk with intrinsic dimension dk ≪ D. In practice, real task distributions (e.g., across language tasks, vision tasks, or embodied tasks) may have overlapping manifolds or varying intrinsic dimensions. How does the theoretical advantage of routing-based Agentic AI degrade as manifold overlap increases? Can you provide bounds in terms of manifold overlap?

**Routing Perfect Assumption**: Proposition 3.2 and subsequent analysis assume routing function π is fixed or introduces "negligible error." How sensitive are the exponential efficiency gains in Equation 3 to routing errors? Can you provide a tighter joint bound that doesn't factor out routing error, or quantify how accuracy requirements on π scale with K and the desired convergence rate?

**Alternative Views Section:**

Yes

**Compliance With Llm Reviewing Policy A Conservative:**

Affirmed.

**Discussion Potential:**

3

**Final Justification:**

My concerns have been solved. I keep my score unchanged as 5 and recommend to accept this paper.

**Paper Summary:**

This position paper argues that Agentic AI uisng multi-agent systems with task decomposition and coordinated autonomy,represents a superior pathway to AGI compared to purely single model scaling.

The authors challenge the prevailing assumption that indefinite scaling of single models is sufficient, instead proposing that heterogeneous task distributions inherent in real-world problems necessitate multi-agent architectures.

Through theoretical derivations grounded in learning theory, the paper establishes that monolithic models suffer unavoidable optimization compromises (termed the "Average Trap"). While routing-based Agentic systems achieve exponential gains in sample and parameter efficiency by decomposing high-dimensional problems into low-dimensional manifold-based sub-tasks.

The analysis extends from simple routing mechanisms to general Directed Acyclic Graphs (DAGs) of agents, introducing topological concepts like the Topology Factor that characterizes how agent composition affects generalization. The authors position current multi-agent system failures as solvable design issues rather than fundamental limitations, and call for research prioritization on Agentic AI optimization as an accessible alternative to expensive monolithic scaling.

**Position:**

Yes

**Position In Title:**

Yes

**Related Work:**

3

**Strengths And Weaknesses:**

## Strengths

- The paper takes a clear, timely position that monolithic scaling is insufficient for AGI, and offers Agentic AI as a theoretically grounded alternative relevant to the ICML community.

- The arguments are well-grounded in established learning theory (No Free Lunch Theorem, Curse of Dimensionality, generalization bounds), with the "Structured Real-World Distribution" formalization providing a useful bridge between intuition and formal analysis.

- The "Average Trap" analysis (Proposition 3.3) and the graduated progression from simple routing to general DAG architectures (Section 4) effectively demonstrate why monolithic models face parameter compromise and how agentic decomposition achieves efficiency gains.

## Weaknesses

- The theoretical analysis relies on strong assumptions (clean manifold separation, perfect routing, well-separated task subspaces per Assumption 3.4) that rarely hold in practice. The paper briefly acknowledges this (Remark B.3) but does not characterize how conclusions degrade when assumptions are violated.

- The paper is purely theoretical with no empirical validation. Even preliminary experiments or quantitative comparisons would strengthen claims about exponential efficiency gains. The conceptual boundary between Agentic AI and existing approaches like mixture-of-experts or modular networks also remains unclear.

- Practical concerns around routing error scaling, composition overhead, and agent orchestration are either underspecified or dismissed. The paper lacks guidance on how to design or optimize DAG topologies in practice, limiting actionability.

- The "Average Trap" framing (Proposition 3.3) overstates its case — the magnitude of the penalty depends on task divergence and curvature, but the paper does not characterize when this constraint is actually binding versus negligible.

**Support:**

3

---

> ### Author Rebuttal · Authors · 2026-03-29
>
> Dear Reviewer VfDj,
>
> We thank the reviewer for the thorough engagement and provide analyses.
> > W1&Q1: ... results rely on strong assumptions... The routing advantage degradation as manifold overlap increases?
>
> **A1:** We let $\gamma = \dim(S_{\text{shared}}) / d_{\max} \in [0, 1]$, decomposing $S_k = S_{\text{shared}} \oplus S_k^{\text{res}}$ with $S_j^{\text{res}} \perp S_k^{\text{res}}$. In the shared subspace, tasks agree on a common optimum (zero penalty); in the residual, full divergence applies. The penalty from Prop. 3.3 becomes: $\epsilon(\gamma) \approx (1 - \gamma) \epsilon_{\text{full}}$
>
> The efficiency ratio generalizes to $N_{\text{R-Agentic}}/N_{\text{mono}} \propto K^{d_{\max}} \epsilon^{(1 - \gamma)(D - d_{\max})}$; the exponent shrinks linearly with $\gamma$ but stays negative for $\gamma < 1$. Connecting to Lem. 3.5: $\Delta_{\max}(K, \gamma) \approx L_{\max} (1 - \gamma) (1 - \frac{1}{\sqrt{K}})$
>
> At $\gamma = 0$, this recovers Eq. 3; at $\gamma > 0$, the advantage degrades smoothly but stays positive for $\forall\gamma<1$; at $\gamma = 1$, it reduces to the monolithic case.
> > W2: It is theoretical with no experiments... The boundary between Agentic AI and MoE is unclear.
>
> **A2:** While theoretical, our formalization captures well-stated phenomena: negative transfer [1,2], gradient conflicts [3,4], and success of sparse MoEs [5] and multi-agent specialization [6,7] validate our predictions.
>
> MoE and Agentic AI differ in three aspects:
> 1. Scope: MoE uses fixed experts with learned gating in one forward pass; Agentic AI uses autonomous agents with independent parameters and multi-step reasoning.
> 2. Topology: MoE is single-layer routing; ours generalizes to arbitrary DAGs (Sec. 4).
> 3. Routing: MoE is differentiable; agentic routing accommodates iterative refinement and external knowledge.
>
> > W3: ... composition overhead and agent orchestration are either underspecified or dismissed. No guidance on DAG optimization.
>
> **A3:** Sec. 4 provides concrete guidance:
>
> 1. Topology via $C(G)$. $C(G) = \sum_{u=1}^K \omega_u$ decomposes into gain ($\prod J_e$) and interference ($\sum_\rho$). Two canonical cases: (1) Pure routing (MoE-like): inherently stable, $C(G) \approx \sum L_u$. (2) CoT (sequential): contractive agents ($\|J\| < 1$) yield exponential error decay; expansive agents ($\|J\| > 1$) cause explosion ($C(G) \propto \lambda_{\max}^K$).
>
> 2. Edge design via $\mathcal{W}(e^*)$. Lem. 4.2 decomposes edge contributions into Upstream History, Local Valve, and Downstream Sensitivity, yielding two rules: (1) After long chains: use contractive edges, like Critic. (2) Before critical decisions: use Verification edges ($\|J\| \ll 1$).
>
> > W4: ... it does not characterize the Average Trap penalty magnitude when this constraint is binding versus negligible.
>
> **A4:** The Average Trap is mild for narrow task families but increasingly binding as the distribution broadens toward AGI. Per Prop. 3.3, $\epsilon = \sum_k \frac{\alpha_k}{2} \|\theta_{\text{mono}}^* - \theta_k^*\|^2_{H_k}$ accumulates with $K$, compounded by expanding $D$ of $\bigcup \mathcal{M}_k$ (Eq. 3). Monolithic models succeed on narrow benchmarks (small $K$) but fail on diverse leaderboards; for AGI-level generality, agentic decomposition becomes necessary.
> > Q2: Sensitivity of exponential efficiency gains in Eq. 3; Need a tighter joint bound that doesn't factor out routing error.
>
> **A5:** Substituting routing error $\epsilon_\pi$ and mismatch $\Delta$ into Eq. 2:
>
> $$\mathcal{E}_{\text{R-Agentic}}(K, N) \leq \underbrace{\frac{K C_{\exp}}{N^{1/d_{\max}}}}_{\text{decreases with } K} + \underbrace{\Delta \epsilon_\pi(K)}_{\text{increases with } K}$$
>
> This yields a U-shaped profile ($K\to 1$: insufficient specialization; $K\to\infty$: routing overwhelms), with optimal $K^*$ in between. And from Sec. 3.2:
>
> Tree-based routers: $\mathcal{E} \leq \frac{K C}{N^{1/d_{\max}}} + C_{\text{tree}} L_{\max} (1 - \frac{1}{\sqrt{K}}) \sqrt{\frac{\text{poly}(\log K)}{N_{\text{router}}}}$. Modularity cost grows polylogarithmically, so specialization dominates for vast $K$.
> Neural routers: $\mathcal{E} \leq \frac{K C}{N^{1/d_{\max}}} + C_{\text{NN}} L_{\max} (1 - \frac{1}{\sqrt{K}}) \sqrt{\frac{K}{N_{\text{router}}}}$. Cost rises as $\sqrt{K}$, restricting $K^*$ unless $N_{\text{router}} \propto K$.
>
> In both cases, specialization ($N^{-1/d_{\max}}$ vs. $N^{-1/D}$) dominates polynomial routing cost for large $N$ since $d_{\max} \ll D$.
>
> We will add those analyses, including partial-overlap, MoE distinction, and joint bound, in the revision and are happy to further discuss these topics with the reviewer.
>
> ---
>
> [1] Which Tasks Should Be Learned Together in Multi-task Learning? Standley et al., ICML'20
>
> [2] Efficiently Identifying Task Groupings for Multi-Task Learning. Fifty et al., NIPS'21
>
> [3] PCGrad, Yu et al., NIPS'20
>
> [4] CAGrad, Liu et al., NIPS'21
>
> [5] Switch transformers, Fedus et al. JMLR'22
>
> [6] MAST, Pan et al., NIPS'25
>
> [7] AgentNet, Yang et al., NIPS'25

---

> > ### Author Rebuttal · Reviewer_VfDj · 2026-04-03
> >
> > My concerns have been solved. As I already gave 5 score, I will not change in this round.

---

### Official Review · Reviewer_7Xwi · 2026-03-12

**Significance:** 3
**Argument Clarity:** 3
**Rating:** 5
**Confidence:** 3

**Questions:**

Please see weaknesses.

**Alternative Views Section:**

Yes

**Compliance With Llm Reviewing Policy A Conservative:**

Affirmed.

**Discussion Potential:**

2

**Paper Summary:**

The paper takes a position on the way of achieving AGI in respect to large monolithic models against agentic systems consisting of multiple task specialized models. The authors take the position that scaling a single model is not enough for AGI and make the case for Agentic AI as a viable path for AGI. Specifically, they take this position with regards to agentic systems built on multi-agent collaboration, dynamic task decomposition, and coordinated autonomy. These systems can adaptively to decompose tasks into correlated subtasks and orchestrate specific agents which are trained with task-specific biases.

The authors base their position on well-developed theoretical study showing the limitation of single model as against multiple agents routed intelligently. They show that routing / DAG topology based agentic model sample complexity is grows slowly relative to monolithic model with dimensionality of the tasks.

**Position:**

Yes

**Position In Title:**

Yes

**Related Work:**

2

**Strengths And Weaknesses:**

### Strengths:
1. The case for monolithic model limitation in terms of learning tasks with non-overlapping parameters is clean.
1. The development of DAG topological agents provides an interesting perspective.

### Weaknesses:
1. The assumption of Task divergence is likely not realistic, that functional distances between tasks implies distinct optimization landscapes. In fact, we know so many tasks share underlying structure and reasonably their function spaces would overlap. In such case, the theory proposed does not hold.


### Overall,
I find the paper takes a recently popular position that Agentic AI is a promising approach for AGI. However, the authors formulate the kind of Agentic AI that is suitable and has less sample complexity compared to monolithic models, in terms of routing and DAG topologies. Their perspective seems to be well supported, although the assumptions are not too simple.

**Support:**

3

---

> ### Author Rebuttal · Authors · 2026-03-29
>
> Dear Reviewer 7Xwi,
>
> We sincerely thank the reviewer for the valuable question and review.
>
> > Weakness: The ass. of Task divergence is likely not realistic, that functional distances between tasks implies distinct optimization landscapes... In such case, the theory proposed does not hold
>
> **Answer**: We sincerely thank the reviewer for the positive assessment and the thoughtful concern regarding the task divergence assumption. We agree that full task divergence is an idealization and in the real world, tasks often share partial underlying structure. Below, we clarify why the paper's theoretical conclusions remain valid and strengthened, under partial overlap.
>
> **1. The theory requires divergence, not independence**
>
> Prop. 3.3 requires that task-optimal parameters are *non-identical* ($\exists\, i,j: \theta_i^* \neq \theta_j^*$), **not** that tasks share no structure. These are fundamentally different conditions. Tasks can share the vast majority of their optimal representations while still inducing a strictly positive Average Trap penalty through their residual differences.
>
> Concretely, considering the penalty term $\epsilon$ from Prop. 3.3, even if the task-optimal parameters $\theta_k^*$ cluster tightly (as they would for structurally related tasks), as long as they do not all coincide, $\epsilon > 0$. The magnitude scales with the divergence, but the *existence* of the penalty is guaranteed by any nonzero heterogeneity.
>
>
> **2. Degradation under partial overlap**
>
> To formalize the reviewer's concern, we introduce an overlap parameter $\gamma \in [0, 1]$ that quantifies the degree of shared structure between task subspaces.
>
> Following Assumption 3.4, each task $k$ occupies a feature subspace $S_k \subset \mathbb{R}^D$. Under partial overlap, we decompose: $S_k = S_{\text{shared}} \oplus S_k^{\text{res}}$, where $S_{\text{shared}}$ is the common subspace across all tasks with $\dim(S_{\text{shared}}) = \gamma \cdot d_{\max}$, and $S_k^{\text{res}}$ is the task-specific residual subspace with $S_j^{\text{res}} \perp S_k^{\text{res}}$ for $j \neq k$.
>
> The penalty from Proposition 3.3 decomposes into shared and residual components. In the shared subspace, all tasks can agree on a common optimum (no penalty). In the residual subspace, the original divergence applies. Therefore:
> $$\epsilon(\gamma) \approx (1 - \gamma) \cdot \epsilon_{\text{full}}$$
> where $\epsilon_{\text{full}}$ is the penalty under full divergence ($\gamma = 0$).
>
> For sample complexity, the effective intrinsic dimension for the agentic system becomes $d_{\text{eff}}(\gamma) = \gamma \cdot d_{\max} + (1 - \gamma) \cdot d_{\max} = d_{\max}$ for the shared component plus a reduced residual. The key sample complexity ratio generalizes to:
> $$\frac{N_{\text{R-Agentic}}}{N_{\text{mono}}} \propto K^{d_{\max}} \cdot \epsilon^{(1-\gamma)(D - d_{\max})}$$
>
> When full separation ($\gamma = 0$), it recovers Equation (3) exactly. And when overlap happens ($\gamma > 0$), the advantage degrades smoothly but remains strictly positive for $\forall\gamma<1$. The extreme case is when complete overlap, the system reduces to the monolithic case.
>
> We appreciate that the reviewer raises this point and commit to formalizing the partial-overlap regime as an extra remark in the revision.
>
> **3. Scaling strengthens the argument**
>
> Crucially, the intuition that shared structure weakens the theory is most applicable for small $K$ (few, closely related tasks). However, the paper's central argument targets **AGI-level task distributions** where $K$ is large. In this regime, even partial divergence becomes increasingly costly:
>
> On the one hand, the accumulated penalty $\epsilon = \sum_k \frac{\alpha_k}{2}\|\Delta_k\|^2_{H_k}$ grows with $K$ as the monolithic optimum must compromise across more divergent directions. On the other hand, the effective ambient dimension $D$ of the union manifold $\bigcup \mathcal{M}_k$ grows with $K$, exacerbating the curse of dimensionality that Equation (3) captures.
>
> Thus, the theory is most relevant precisely in the regime the paper targets: broad, heterogeneous task distributions approaching general intelligence. In this limit, scaling a monolithic model forces an increasingly severe compromise across orthogonal residual dimensions, making Agentic AI not just an alternative, but a structural necessity for AGI.
>
> We appreciate these reviews, which help us make the paper more complete, and are very happy to further discuss these topics with the reviewer during the discussion phase.

---

> > ### Author Rebuttal · Reviewer_7Xwi · 2026-04-04
> >
> > Thanks for the response. I maintain my score.

---

### Official Review · Reviewer_QDAL · 2026-03-12

**Significance:** 4
**Argument Clarity:** 3
**Rating:** 5
**Confidence:** 4

**Questions:**

While this paper emphasizes task decomposition and agent specialization based on the assumption that real-world tasks reside on distinct manifolds, a counter-argument exists. In many practical scenarios, diverse tasks exhibit significant representation overlap. Monolithic models often enhance generalization by learning these shared representations. A prime example is recent Multimodal Large Language Models (MLLMs), which demonstrate robust generalization across text, images, and video by mapping different modalities into a single, unified representation space.

Given this context, what are the grounds for asserting that Agentic AI remains superior to monolithic models?

**Alternative Views Section:**

Yes

**Compliance With Llm Reviewing Policy A Conservative:**

Affirmed.

**Discussion Potential:**

4

**Final Justification:**

Thank you for the clear and constructive rebuttal. The clarification on the compatibility between shared representations and agentic decomposition, along with the introduction of the overlap parameter, adequately addresses my concerns. I will update my score accordingly.

**Paper Summary:**

This paper argues that monolithic scaling is insufficient for achieving Artificial General Intelligence (AGI). Instead, it posits that an Agentic AI architecture, characterized by the collaboration of multiple agents, represents a more realistic path toward AGI capable of handling the complexities of the physical world.
To support this claim, the paper presents three primary arguments:
1. The "Average Trap" in Real-World Distributions
By analyzing the structured real-world distribution, the paper demonstrates that real-world problems manifest as a collection of heterogeneous tasks composed of different manifolds and functions. In such environments, a single model must optimize for all tasks simultaneously, leading to conflicts between optimal solutions. This results in the "Average Trap," where the model's performance converges to a mediocre average across all domains.

2. Sample Efficiency and Generalization
The paper provides a theoretical analysis showing that routing-based Agentic architectures offer superior sample efficiency and generalization compared to monolithic models. While the generalization error of a single model is constrained by the input dimension D, Agentic AI decomposes problems into sub-tasks. This allows each agent to learn within a lower intrinsic dimension, leading to a significantly faster reduction in generalization error.

3. DAG-based Modeling and Convergence
The paper models a generalized Agentic AI as a system where agents are interconnected via a Directed Acyclic Graph (DAG). It proves that the total error of the system is determined by the local error of each agent and a topological weight defined by the graph's topology. Theoretically, this explains how a well-designed Agentic system can achieve faster convergence and more efficient learning than a monolithic model.

In conclusion, the paper asserts that achieving AGI requires a paradigm shift: moving away from simple model scaling and toward an Agentic AI framework that leverages task decomposition and multi-agent collaboration.

**Position:**

Yes

**Position In Title:**

Yes

**Related Work:**

2

**Strengths And Weaknesses:**

### Strengths
- Novel Theoretical Foundation for the Main Argument:
The paper offers a compelling attempt to model real-world task distributions as manifold structures and explains the limitations of monolithic models through the lenses of the curse of dimensionality and sample complexity.

- Addressing a Pivotal Debate in Current AI Research:
The tension between monolithic scaling and agentic/multi-agent architectures as the path to AGI is a central theme in modern AI. This paper meaningfully contributes by attempting to frame this techno-philosophical debate within a rigorous theoretical context.

- Formalization of Agentic AI as a DAG-based Compositional System:
Defining multi-agent collaboration as a Directed Acyclic Graph (DAG) where the topology directly influences system performance is a significant contribution. This provides a structured framework for future system design and analysis of information flow between agents.

### Weaknesses
- Insufficient Rebuttal of Shared Representation Learning:
One of the strongest arguments for monolithic scaling is that tasks are rarely independent; they often share a common underlying representation that enhances generalization. The paper focuses heavily on the structural differences between tasks but fails to sufficiently address or counter the advantages of shared representation learning inherent in large-scale monolithic models.

- Overly Rigid Theoretical Assumptions:
The assumptions that tasks exist on entirely distinct manifolds or that subspaces are nearly orthogonal simplify the mathematical analysis but may not reflect reality. In practical scenarios, task representations often overlap significantly, a condition under which the strengths of a monolithic model might actually be more pronounced.

**Support:**

2

---

> ### Author Rebuttal · Authors · 2026-03-29
>
> Dear Reviewer QDAL,
>
> We sincerely thank the reviewer for the valuable review. Our central response is that agentic decomposition and shared representations are compatible, not contradictory.
>
> > W1: ... One of the arguments for monolithic scaling is that tasks share a common underlying representation that enhances generalization.
>
> **A1:** We wish to clarify that the paper does not argue against shared representations; rather, we formalize their performance ceiling under heterogeneous task distributions.
>
> Agentic AI actually relies on shared representations, like embeddings. Agentic decomposition handles the **residual** task-specific components that a shared backbone cannot capture. This mirrors the MoE pattern, where a shared backbone captures common structure while routed experts handle task-specific divergence.
>
> Formally, shared representations reduce, but cannot eliminate, the penalty. From Prop. 3.3: $\epsilon = \sum_{k=1}^{K} \frac{\alpha_k}{2} \|\theta_{\text{mono}}^* - \theta_k^*\|^2_{H_k}$
>
> This penalty is strictly positive ($\epsilon > 0$) whenever optimal parameters differ between any two tasks ($\exists\, i,j: \theta_i^* \neq \theta_j^*$). Even if tasks share 99% of their optimal representation, the remaining 1% divergence accumulates. Especially, for AGI-level distributions where $K$ is extremely large, the sum cannot be ignored.
>
> Additionally, empirical evidence supports this analysis. Jointly training related tasks frequently degrades performance [1], and gradient conflicts between tasks are pervasive [2,3], which directly supports the mechanism behind Prop. 3.3. And the established success of MoE architectures proves that routing to specialized sub-networks improves performance even when tasks share a common backbone [4,5].
>
> > W2: ..., Ass. 3.4 may not reflect reality. Under huge overlap, monolithic models might actually be stronger.
>
> **A2:** To better explain the overlap impact, we let $\gamma = \dim(S_{\text{shared}}) / d_{\max} \in [0, 1]$ to quantify the degree of shared representation. Each task's feature subspace decomposes as: $S_k = S_{\text{shared}} \oplus S_k^{\text{res}}$
>
> where $S_{\text{shared}}$ is common across all tasks and $S_k^{\text{res}}$ captures task-specific residuals with $S_j^{\text{res}} \perp S_k^{\text{res}}$ for $j \neq k$.
>
> In the shared subspace, all tasks agree on a common optimum (zero penalty). In the residual subspace, the full divergence applies. Then penalty from Prop. 3.3 becomes: $\epsilon(\gamma) \approx (1 - \gamma) \cdot \epsilon_{\text{full}}$
>
> For sample complexity, the efficiency ratio generalizes to:
> $\frac{N_{\text{R-Agentic}}}{N_{\text{mono}}} \propto K^{d_{\max}} \cdot \epsilon^{(1 - \gamma)(D - d_{\max})}$
> The exponent shrinks linearly with $\gamma$ but remains strictly negative for any $\gamma < 1$, preserving the exponential advantage.
>
> And for mismatch penalty, connecting to Lem. 3.5, the mismatch penalty becomes:
> $\Delta_{\max}(K, \gamma) \approx L_{\max} \cdot (1 - \gamma) \cdot (1 - \frac{1}{\sqrt{K}})$
> When full separation ($\gamma = 0$), it recovers Eq. 3 exactly. And when overlap happens ($\gamma > 0$), the advantage degrades smoothly but remains strictly positive for $\forall\gamma<1$. The extreme case is when complete overlap, the system reduces to the monolithic case.
>
> We frame Ass. 3.4 as a best-case analysis establishing the ceiling of agentic advantage. We commit to formalizing the partial-overlap regime as an extra remark in the revision.
>
> > Q1: MLLMs demonstrate robust generalization by mapping different modalities into a single unified representation space. Why is Agentic AI still superior?
>
> **A3:** We'd like to address this on four levels.
>
> (1) MLLMs and Agentic AI are complementary, not competing. The true comparison is a single monolithic MLLM versus specialized MLLMs orchestrated agentically.
>
> (2) MLLMs still underperform fine-tuned specialists on domain-specific tasks. As Prop. 3.3 predicts, unified models trade expert-level acuity for broad usability.
>
> (3) Modality $\neq$ Task. A shared input representation does not resolve conflicting optimization objectives for functionally diverse tasks. Task divergence in Def. 2.1 concerns differing optimal labeling functions ($f_k$), not merely input encodings.
>
> (4) Modern MLLMs increasingly rely on agentic patterns like tool calling and multi-step reasoning [6]. This demonstrates that even monolithic models are evolving toward the decomposed, routed paradigm our theory recommends.
>
> We appreciate these reviews help us make the paper more complete and are very happy to further discuss these topics with the reviewer during the discussion phase.
>
> ---
> [1] Which Tasks Should Be Learned Together in Multi-task Learning? Standley et al., ICML'20
>
> [2] PCGrad, Yu et al., NIPS'20
>
> [3] CAGrad, Liu et al., NIPS'21
>
> [4] Switch transformers, Fedus et al. JMLR'22
>
> [5] GShard, Lepikhin et al., ICLR'21
>
> [6] AI Agents vs. Agentic AI, Sapkota et al., Information Fusion'26

---

> > ### Author Rebuttal · Reviewer_QDAL · 2026-04-04
> >
> > Thank you for the clear and constructive rebuttal. I will update my score accordingly.

---

### Decision · Program_Chairs · 2026-04-30

**Decision:**

Accept (regular)

**Comment:**

Reviewers all praised the paper for tackling a central debate in the current AI discourse, and for connecting this debate to rigorous formalisms within machine learning. Reviewers especially appreciated the DAG-based view of agents.

In light of the positive appraisal by the reviewers, I recommend acceptance. I encourage the authors to take the reviewers' comments into account in their revisions.